# AQuaRef: machine learning accelerated quantum refinement of protein structures

Roman Zubatyuk[1,7], Malgorzata Biczysko [2,7], Kavindri Ranasinghe[3,7], Nigel W. Moriarty [4], Hatice Gokcan[1], Holger Kruse[5], Billy K. Poon[4], Paul D. Adams [4,6], Mark P. Waller[5], Adrian E. Roitberg[3], Olexandr Isayev [1] ✉ & Pavel V. Afonine[4] ✉

Cryo-EM and X-ray crystallography provide crucial experimental data for obtaining atomic-detail models of biomacromolecules. Refining these models relies on library-based stereochemical data, which, in addition to being limited to known chemical entities, do not include meaningful noncovalent interactions. Quantum mechanical (QM) calculations could alleviate these issues but are too expensive for large molecules. Here we present a novel AI-enabled Quantum Refinement (AQuaRef) based on AIMNet2 machine learned interatomic potential (MLIP) mimicking QM at substantially lower computational costs. By refining 41 cryo-EM and 30 X-ray structures, we show that this approach yields atomic models with superior geometric quality compared to standard techniques, while maintaining an equal or better fit to experimental data. Notably, AQuaRef aids in determining proton positions, as illustrated in the challenging case of short hydrogen bonds in the parkinsonism-associated human protein DJ-1 and its bacterial homolog YajL.

While advances in predictive modeling, such as AlphaFold3[1] or RoseTTAFold[2,3], have provided powerful tools for structural biology, they remain limited while experimental methods, including protein crystallography and cryo-EM, are still cornerstones of structural biology and drug development[4]. Experimental data allow for the discovery of new structures emerging in life evolution, potentially exhibiting previously unseen features. These discoveries require unbiased information provided by experiments to explore the unknown[5]. Atomic model refinement is a crucial near-final stage in crystallographic or cryo-EM structure determination aimed at producing molecular models that meet standard validation criteria while optimally fitting the experimental data[6,7]. Refinement heavily relies on stereochemical restraints to maintain the correct geometry of the atomic model while fitting to the experimental data[8]. These restraints originate from standard libraries that tabulate the topology and parameters of known chemical entities[9,10], which are universally

employed across popular software packages, such as CCP4[11] and Phenix[12].

The limitations of library-based restraints are manifold. Firstly, they only include terms for maintaining covalent bond lengths, bond angles, torsion angles, planes, and chirality while preventing clashes through non-bonded repulsion[13]. However, it has been demonstrated that at low resolution, these restraints are insufficient to maintain realistic, chemically meaningful macromolecular geometries, making it essential to include additional restraints on protein main chain $\varphi/\psi$ angles, side chain torsion $\chi$ angles, as well as hydrogen bond parameters and π-stacking interactions to stabilize protein or nucleic acid secondary structure[13–19]. These additional restraints cannot be reliably inferred from the atomic model alone and thus require manual error-prone annotation and curation using additional sources of information, such as homologous high-resolution models. Secondly, library-based restraints parametrize only known chemical entities, such as

[1]Department of Chemistry, Carnegie Mellon University, Pittsburgh, PA, USA. [2]Faculty of Chemistry, University of Wrocław, Wrocław, Poland. [3]Department of Chemistry, University of Florida, Gainesville, FL, USA. [4]Molecular Biophysics & Integrated Bioimaging Division, Lawrence Berkeley National Laboratory, Berkeley, CA, USA. [5]Pending.AI, Eveleigh, NSW, Australia. [6]Department of Bioengineering, University of California Berkeley, Berkeley, CA, USA. [7]These authors contributed equally: Roman Zubatyuk, Malgorzata Biczysko, Kavindri Ranasinghe. ✉e-mail: olexandr@olexandrisayev.com; PAfonine@LBL.Gov

standard amino and nucleic acids, as well as previously defined ligands. Consequently, any nonstandard entities or interactions, such as novel ligands or covalent cross-chain links, require manual annotation and definition, without which refinement may fail to proceed correctly or at all. Finally, deviations from standard covalent geometry due to local chemical interactions are not uncommon[20–22]. While these deviations are valid, restraints may interpret them as violations requiring 'correction'.

The advantage of using simple restraints[8] is the minimal computational cost they add to the refinement workflow. A possible next step is to use a classical force field to account for geometric elements[23]. However, these force fields have their own set of limitations: they require parametrization for new chemical species and cannot distinguish between chemically equivalent bonds in different chemical environments.

Quantum refinement is a fundamentally different approach, balancing the fitting to experimental data with a term related to the quantum mechanical energy of the system[24,25]. It has been demonstrated that the entire atomic model can benefit from a full QM treatment[26–28]. Figure 1 presents a timeline showcasing the evolution of quantum mechanics calculations for proteins. It highlights four key stages of progress and advancements in technology and methodology, particularly those required for model refinement, where energy and gradients are evaluated for the entire protein structure hundreds to thousands of times. Traditionally, quantum refinements were deemed impractical for macromolecules due to the computational requirements. Methods often focused solely on the macromolecular region of interest, such as a ligand-binding pocket or enzyme active site, while employing a classical approach for the rest of the molecule[29–32]. Numerous approaches and implementations have been reported over time[33], with GPU-accelerated codes enabling QM calculations for peptides and small proteins of a few hundred atoms being one of the most prominent milestones[34]. Interaction-based model partitioning into chemically meaningful fragments[35] solved the scalability issue in quantum calculations[26], which in turn enabled the refinement of larger proteins. However, this approach remained computationally demanding.

Here, we show that Machine Learning Interatomic Potentials (MLIPs) offer a computationally tractable alternative to full quantum refinement. We introduce AI-enabled Quantum Refinement (AQua-Ref), which employs a specialized potential developed using the AIMNet2 architecture[36]. To tailor this potential for structural refinement, we trained a model (see Methods) on a custom-developed dataset for polypeptides that incorporates an implicit solvent correction. This approach leverages the high computational efficiency of the AIMNet2 architecture, including its rotation-invariant learnable features, message-passing scheme for many-body interactions, and explicit handling of the total system charge. The resulting specialized potential allows AQuaRef to achieve quantum-level fidelity for high-accuracy structural refinement at a fraction of the computational cost. Refinement of selected cryo-EM and X-ray atomic models across various resolutions demonstrates the AQuaRef's ability to produce atomic models with superior geometric quality compared to conventional techniques while maintaining or improving agreement with experimental data. This work represents the first example where an MLIP has been adopted to perform quantum refinement of the entire protein, in contrast with a recent approach where ML potentials have been combined with the multi-layer ONIOM-type QM/MM partitioning[37].

## Results

Conceptually, quantum-based atomic model refinement is very similar to classic refinement wherein atomic model parameters are iteratively adjusted in order to minimize the residual, $T = T_{data} + w * T_{restraints}$. Here, $T_{data}$ describes the fit of the model to the data, and $T_{restraints}$ incorporates chemical restraints with an a-priori unknown weight, $w$[38].

However, there are four fundamental differences. First, in QM refinement, restraints are derived from quantum-mechanical calculations for the specific macromolecule in consideration. Second, the requirements for the initial atomic model in QM refinement are stricter compared to standard refinement: the atomic model must be correctly protonated, atom-complete, and free of severe geometric violations such as steric clashes or broken covalent bonds[24,25]. Third, while crystallographic software packages inherently account for crystal symmetry, QM codes generally do not. Fourth, crystallographic software is capable of handling static disorder that is modeled with alternative conformations, whereas QM codes typically lack this capability. All these nuances specific to quantum refinement (except handling static disorder, which is a current limitation) are addressed in the Quantum Refinement package (Q|R)[24,26–28,39], which is being developed as part of this work and provides the necessary procedures to enable quantum refinement within the Phenix software.

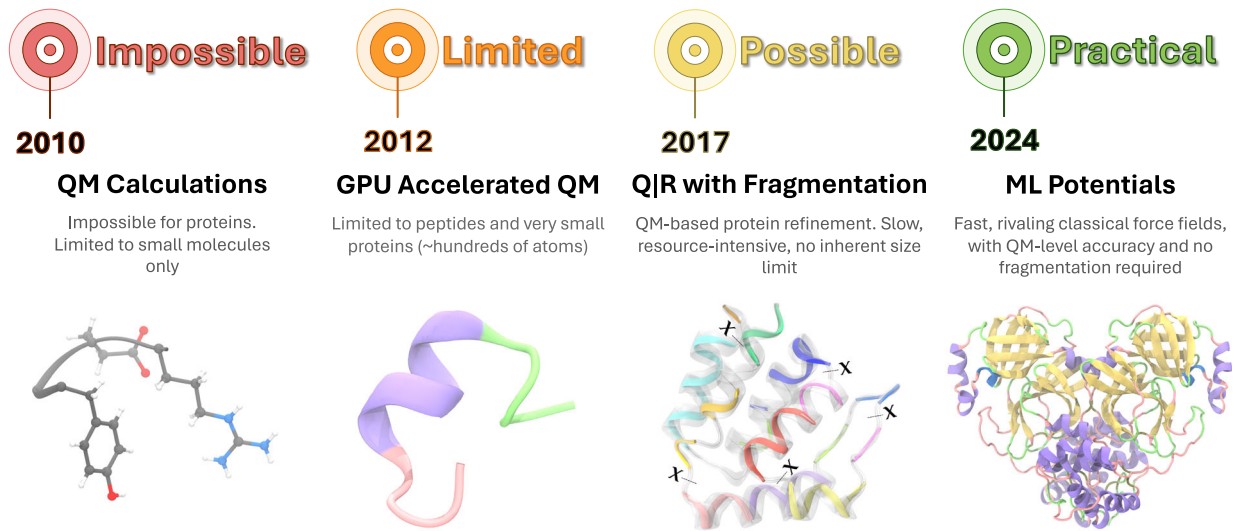

**Fig. 1 | Selected snapshots of progress towards QM-based protein model refinement.** Timeline highlighting selected key milestones in the application of quantum mechanical calculations for refining atomic models of entire protein structures using experimental crystallography or cryo-EM data.

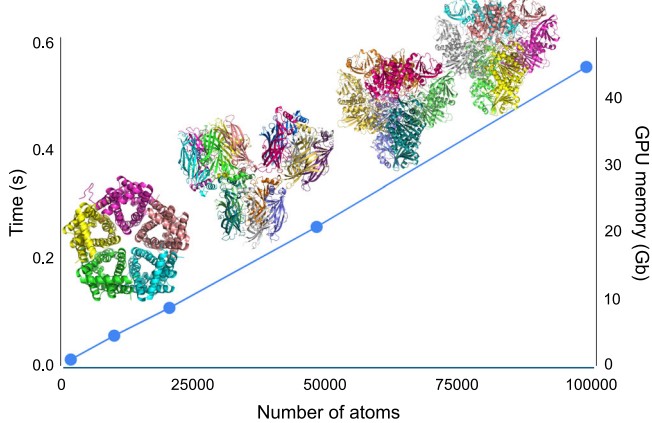

**Fig. 2 | Computational scaling of the AIMNet2 MLIP model in AQuaRef.** Time to compute energy and forces (left axis) and peak GPU memory usage (right axis) versus the number of atoms in the system. Calculations are performed on a single Nvidia H100 PCIE 80GB GPU.

Conventional QM methods like density functional theory (DFT) for $N$-electron systems require $O(N^2)$ storage and $O(N^3)$ arithmetic operations, where $O(\cdot)$ denotes how computational cost scales with system size. This $O(N^3)$ complexity is a critical bottleneck that limits the ability to study large realistic biological systems like proteins. Figure 2 shows the computational scaling of the AIMNet2 model, where both energy and force calculations, as well as peak GPU memory usage, scale linearly ($O(N)$) with system size. For a large protein system of 100,000 atoms, single-point energy and forces can be computed in 0.5 s. Overall, an atomic model consisting of approximately 180,000 atoms can fit into the 80GB memory of a single NVIDIA H100 GPU.

We tested the new quantum refinement procedure on 41 cryo-EM atomic models, 20 low-resolution, and 10 very high-resolution X-ray atomic models. Standard stereochemistry[40,41] and model-to-data fit criteria[42–44], MolProbity validation tools[45], along with newly developed metrics to evaluate hydrogen bond quality[19] were used to assess the atomic models. Typically, the time needed for quantum refinement is about twice as long as standard refinement, and often shorter than the standard refinement with additional restraints such as the Ramachandran plot, secondary structure, and side-chain rotamer restraints[46–48]. Quantum refinement takes under 20 min in about 70% of models considered in this work, with a maximum of about 1 h (Supplementary Table 1). These computations can be performed on GPU-equipped laptops, with the only limitation being available GPU memory.

## Quantum refinement

The AQuaRef refinement procedure begins with a check for the completeness of the atomic model, followed by the addition of any missing atoms. This may result in steric clashes, particularly if the model was previously refined without hydrogen atoms. Models with missing atoms that cannot be trivially added (e.g., missing main chain atoms) cannot be used for quantum refinement. If clashes or other severe geometric violations are detected, quick geometry regularization is performed using standard restraints, ensuring that atoms move as little as necessary to resolve the clashes. For crystallographic refinement, to account for interactions arising from crystallographic symmetry and periodicity of unit cells, the model is expanded into a supercell by applying appropriate space group symmetry operators[27]. Subsequently, it is truncated to retain only parts of the symmetry copies within a prescribed distance from atoms of the main copy[39]. This step is unnecessary for refinement against cryo-EM data. The atom-completed and expanded model then undergoes the standard atomic model refinement protocol as implemented in Q|R package[24].

## Application of the new refinement procedure to a set of deposited atomic models

To evaluate the performance of the new QM-based refinement, we refined 41 low-resolution cryo-EM atomic models, 20 low-resolution, and 10 ultra-high-resolution X-ray atomic models, which contain only proteins. All selected 61 low-resolution atomic models have high-resolution homologs, which were used as the ground truth for comparison (Supplementary Tables 2 and 3). Refinements were carried out using three sets of restraints: QM restraints from AIMNet2 (AQuaRef refinement); standard restraints; and standard restraints plus additional restraints on hydrogen bonds and angles involved in maintaining secondary structure, main-chain $\varphi/\psi$ angles (Ramachandran plot restraints), and side-chain torsion $\chi$ angles (rotamer restraints).

Overall, low-resolution atomic models after quantum refinement exhibit systematically superior geometry quality compared to those obtained using standard restraints, as indicated by their MolProbity scores[49], Ramachandran $Z$-scores[50], CaBLAM disfavored[45] (Fig. 3a), and skew-kurtosis plots for hydrogen bond parameters[19] (Fig. 3d). They also systematically deviate more from the initial coordinates. These atomic models demonstrate a very similar fit to the experimental data (Fig. 3b, c), with slightly less data overfitting for X-ray atomic models, as evidenced by a smaller $R_{work}$-$R_{free}$ gap and similar $R_{free}$[51,52]. Since there is no equally efficient control over overfitting in cryo-EM as there is with $R_{free}$ in crystallography, the slightly lower cross-correlation between experimental and model-calculated masked maps ($CC_{mask}$)[42] and essentially the same EMRinger scores[53], together with significantly improved atomic model geometry, likely indicate a reduction in overfitting. Augmenting standard restraints with secondary structure, Ramachandran plot, and side-chain rotamer restraints expectedly improves the geometry (Fig. 3d, f), yet using AQuaRef still produces superior atomic model geometries. With a few exceptions, atomic models refined with quantum restraints are systematically closer to their higher-resolution homologs compared to those using standard restraints alone or complemented with additional restraints (Fig. 3e, f). In some of the most remarkable cases, the local structure obtained with AQuaRef restraints closely matches the high-resolution homologs and differs from those obtained using standard restraints by up to two Angstroms (Fig. 4).

## Comparison with alternative state-of-the-art approaches

To further evaluate the performance of AQuaRef refinement compared to other major refinement methods and software, we refined selected low-resolution X-ray models using the AMBER force field as a source of geometric restraints[23], the Rosetta all-atom force field combined with its powerful sampling methods[54], and standard refinement as implemented in REFMAC5[55]. For cryo-EM, there are fewer refinement alternatives, with Servalcat[56] being the most popular, which we also used in this analysis.

For X-ray models, AQuaRef produced slightly better overall $R_{free}$ values (Fig. 5a) and substantially less data overfitting, as indicated by the $R_{free}$-$R_{work}$ gap (Fig. 5b). For cryo-EM models Servalcat lead to notably better $CC_{mask}$ (Fig. 5c) and both scored the same by EMRinger method (Fig. 5d). Models refined using AQuaRef and Rosetta performed similarly well in terms of Rama-$Z$ scores, achieving excellent results in most cases, while REFMAC5 and Servalcat had the worst scores, and AMBER fell somewhere in between (Fig. 5e). In terms of MolProbity scores and CaBLAM outliers (Fig. 5f, g), AQuaRef and Rosetta also performed similarly well, significantly outperforming REFMAC5 and Servalcat. Rosetta-refined models were closest to the high-resolution reference models, followed by AQuaRef (Fig. 5h). This is likely due to Rosetta's use of non-gradient optimization techniques, such as sampling and local model repacking, which have a larger convergence radius compared to the gradient-driven minimization used in other programs. Finally, AQuaRef and Rosetta both produced models that fit the expected distribution of hydrogen bond parameters (Figs. 5i and 3d), followed by AMBER. REFMAC5 and Servalcat

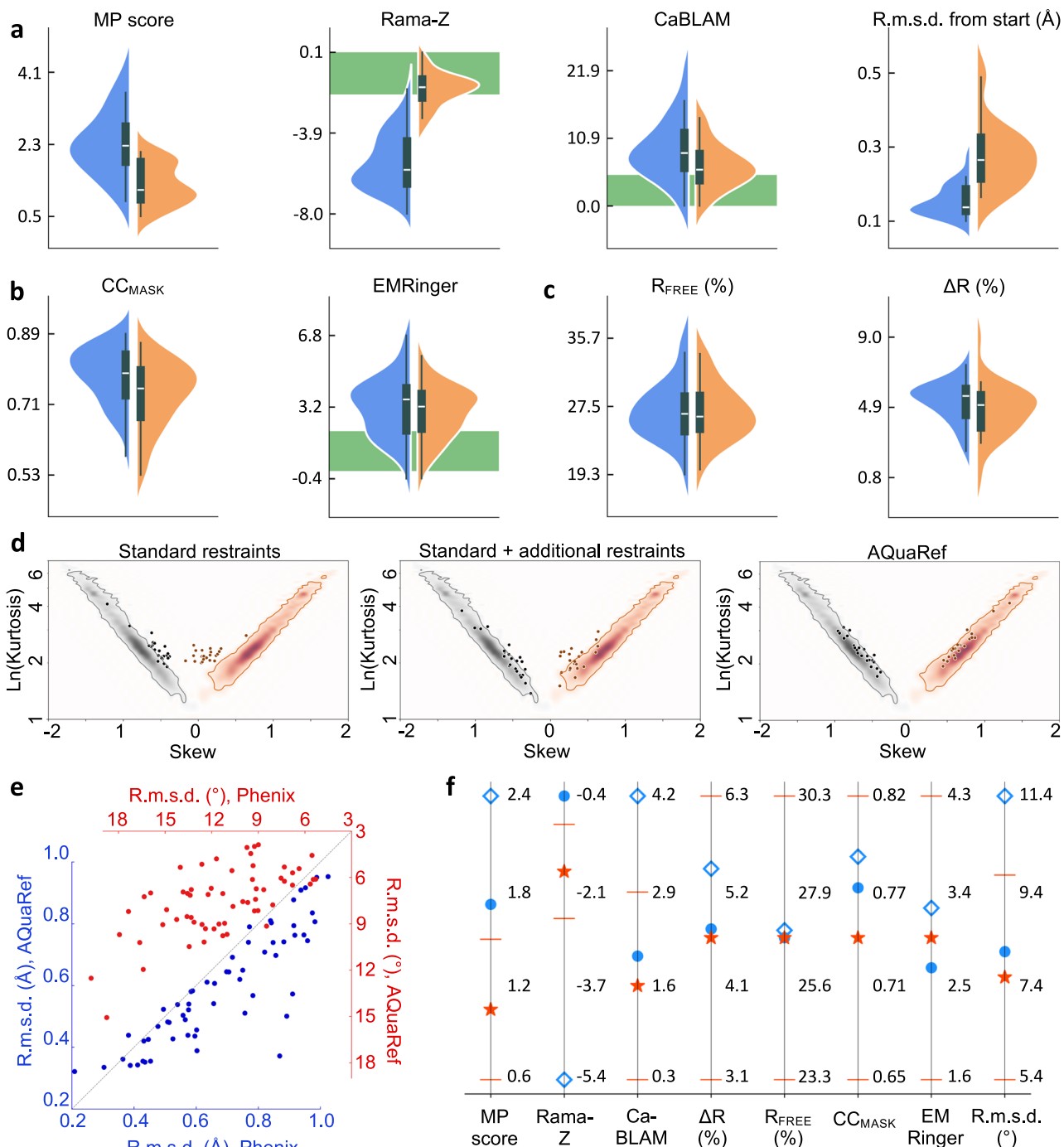

**Fig. 3 | Summary of refinements for 41 cryo-EM models and 20 X-ray models, both at low resolution, using classic approach as implemented in Phenix and AIMNet2 MLIP model in AQuaRef. a–c** Refinements using standard stereo-chemistry (blue) and AQuaRef (orange) restraints. In all box plots the center line indicates the median, the lower and upper bounds correspond to the 25th and 75th percentiles, respectively. The whiskers extend to encompass values that are within 1.5 times the interquartile range from the lower and upper quartiles. **a** MolProbity (MP) score, Ramachandran plot Z-score (Rama-Z), CaBLAM disfavored and r.m.s. deviation of refined model from initial model. **b** cross-correlation between experimental and model-generated maps ($CC_{mask}$), and EMRinger score for cryo-EM models. **c** $R_{free}$ and $R_{free}$-$R_{work}$ (ΔR) for X-ray models. Green band indicates favored range of corresponding values. **d** skew-kurtosis plots for hydrogen bond parameters (Hydrogen(H)…Acceptor(A) distances and Donor-H…A angles) for refinements using (left-to-right): standard restraints; standard restraints with addition of Ramachandran plot, secondary-structure and side-chain rotamer

restraints; and AQuaRef restraints. **e** r.m.s. deviations between refined and high-resolution homology models, refinements using standard versus AQuaRef restraints, calculated using matching Cartesian coordinates (blue, lower-left) and matching torsion angles (red, upper-right). **f** summary of mean values, for all test refined models: MolProbity score, Ramachandran Z-score, CaBLAM outliers, r.m.s. deviation of matching torsion angles between refined and high-resolution homologous models, as well as $R_{free}$-$R_{work}$ (ΔR) and $R_{free}$ for X-ray models and $CC_{mask}$ and EMRinger score for cryo-EM models for refined models with standard restraints (blue rhombi), standard restraints with addition of Ramachandran plot, secondary-structure and side-chain rotamer restraints (blue circles); and AQuaRef restraints (red stars). Red bars show standard deviations for starred values. For geometric model quality measures (MolProbity score, Rama-Z, CaBLAM, and r.m.s. deviations between refined and homology models), the mean and standard deviation were computed using all 61 models. The mean and standard deviation for $CC_{mask}$ were calculated using 41 cryo-EM models, and for $R_{free}$ and ΔR, using 20 X-ray models.

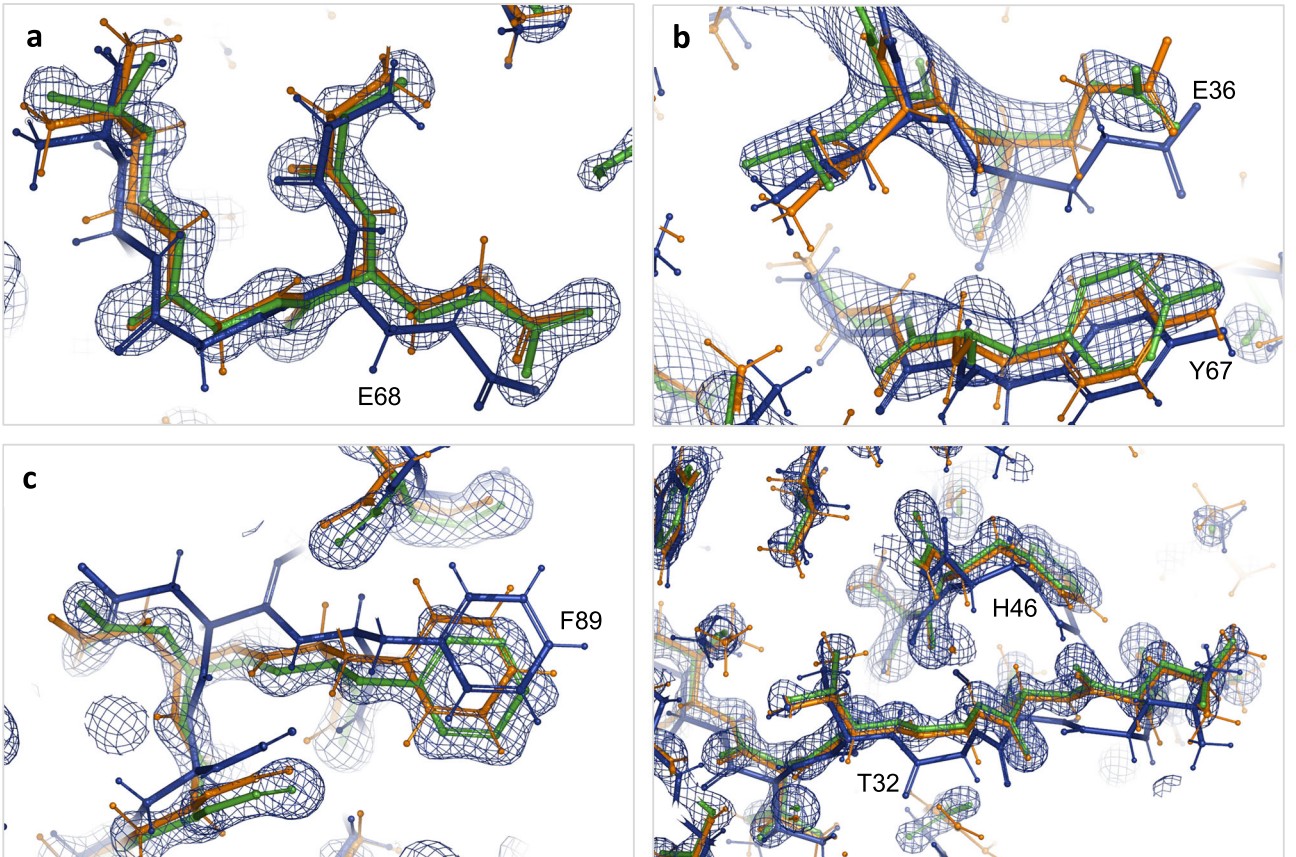

**Fig. 4 | Comparison of low-resolution refined models with high-resolution homologues. a–c** Close-up showing models refined with standard restraints (blue) and AQuaRef restraints (orange) superposed onto their higher-resolution homologous models (green) with their corresponding 2mFo-DFc Fourier maps contoured at 2σ; for PDB 5YI5, 8R1G, and 6XMX, respectively.

largely failed to produce models fitting this distribution, with Servalcat performing the worst.

## Case study: short hydrogen bonds in human DJ-1 and its bacterial homologue YajL

Short hydrogen bonds play a key functional role in proteins, and determining the protonation states of involved residues is critical. However, accurate location of proton positions experimentally remains challenging at resolutions near 1 Å. Lin et al.[57] analyzed high-resolution X-ray crystal structures of human DJ-1 and its bacterial homolog YajL to determine the protonation states of carboxylic acids involved in dimer-spanning hydrogen bonds. Their approach combined bond length analysis, leveraging the distinct lengths of C=O and C−OH bonds, with qualitative interpretation of difference map peaks to identify potential evidence of protons.

This method is complicated by stereochemical restraints applied during coordinate refinement, which can bias bond lengths. For example, in E/D residues, bond length restraints for COOH groups depend on whether a hydrogen atom is explicitly modeled (Fig. 6c). To minimize this bias, Lin et al. performed final rounds of conjugate gradient least-squares refinement in SHELXL[58] without applying restraints to the residues of interest. In contrast, QM-based AQuaRef refinement avoids such biases entirely.

AQuaRef refinement of DJ-1, starting with all E15/D24 CO bond lengths as in an unprotonated state and a single proton symmetrically placed between Oε2 (E15) and Oδ2 (D24) (Fig. 6b), produced proton positions and bond geometries (Fig. 6f) consistent with Lin et al.'s findings (Fig. 6a) and unrestrained refinement using phenix.refine (Fig. 6d). However, restrained refinement with phenix.refine (Fig. 6e)

yielded bond geometries that matched library values assuming no proton on either COO group, highlighting the impact of restraint bias.

To test the robustness of AIMNet2 restraints in preserving accurate geometries, the same refinements were performed using experimental data truncated at 2 Å resolution. This truncation removed atomic-level details that could resolve bond lengths and hydrogen positions. AQuaRef produced results nearly identical to those obtained using the original 1.15 Å atomic resolution data, whereas restrained refinements with phenix.refine further biased oxygen-carbon distances toward idealized values for the unprotonated state (Fig. 6e, f, values in parentheses).

Starting from an idealized symmetric arrangement (Fig. 6b), the refinement could, in principle, place the proton on either E15 or D24. To explain why the proton ultimately settled on Oδ2 of D24, two independent sources of evidence were considered. First, sampling the hydrogen position along the Oδ2−Oε2 bond vector and computing the AIMNet2 energy profile revealed a slight preference for D24 protonation (Fig. 7a). Second, while the resolution and R-factors were insufficient for definitive proton identification in the difference map, the difference map values along the Oε2-Oδ2 axis showed elevated positive values near Oδ2, close to the prospective hydrogen position (Fig. 8a). This, together with the energetic preference from AIMNet2, may have guided the refinement to move the hydrogen toward D24.

Bacterial DJ-1 homologue, the YajL structure, contains two copies of the molecule in the asymmetric unit, resulting in two instances of the E14/D23 interaction. Similar to DJ-1, unrestrained refinement of YajL (Fig. 9b) yielded results consistent with Lin et al. (Fig. 9a). As with DJ-1, restrained refinement introduced significant bias in bond lengths (Fig. 9c) for both instances of the E14/D23 interaction.

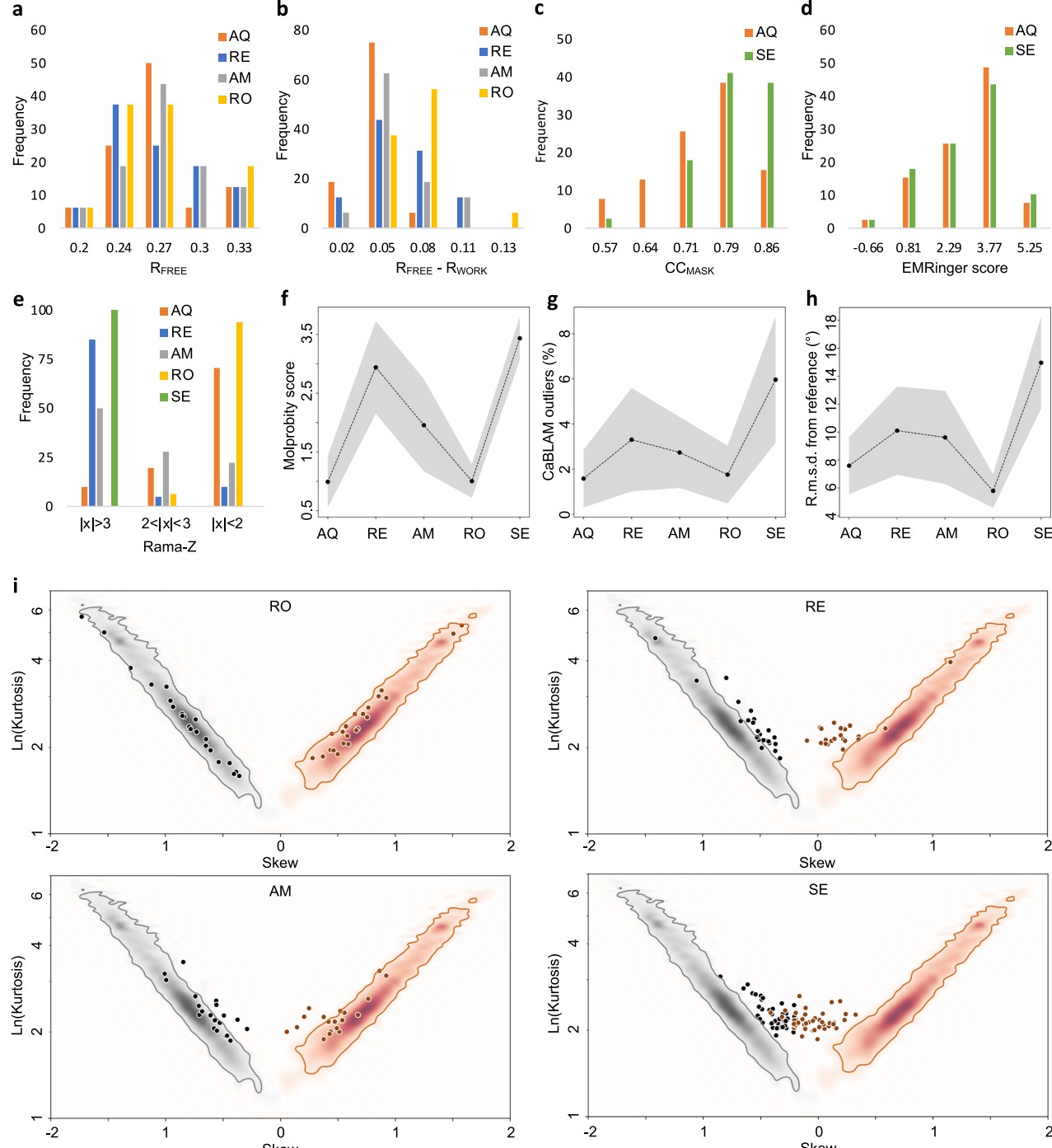

**Fig. 5 | Summary of refinements for 41 low-resolution cryo-EM models using AQuaRef (AQ) and Servalcat (SE), and 20 low-resolution X-ray models using AQuaRef, REFMAC5 (RE), AMBER (AM), and Rosetta (RO). a–e** Distributions of $R_{free}$, $R_{free}$-$R_{work}$, $CC_{mask}$, EMRinger score, and Rama-$Z$, respectively. **f–h** Mean values of MolProbity score, CaBLAM outliers, and r.m.s. deviation from the reference model, calculated across all refined models; gray bands represent the standard deviation. **i** Skew-kurtosis plots for hydrogen bond parameters (Hydrogen(H)…Acceptor(A) distances and Donor-H…A angles) for refinements performed using REFMAC5, AMBER, Rosetta and Servalcat.

Results from AQuaRef refinement aligned with Lin et al. and unrestrained phenix.refine refinement, suggesting that a proton is shared between D23 and E14, rather than being fully localized on either residue. In contrast to DJ-1, the proton in YajL does not appear to be fully associated with one oxygen atom, but rather shared between O$\varepsilon$2 and O$\delta$2, consistent with a Low Barrier Hydrogen Bond (LBHB).

The AIMNet2 energy profile between O$\varepsilon$2 and O$\delta$2 supports this interpretation, showing a relatively flat energy landscape (Fig. 7b). This

indicates that the hydrogen's position could be entirely guided by the experimental data while staying within the flat region of the AIMNet2 energy well. Indeed, there is a significant difference map peaks above 3 s.d. and well above mean solvent density of 0.25 e/Å³, very close to the position of hydrogens in the refined model in both instances of the E14/D23 interaction (Fig. 8b).

Further evidence that C-O$\delta$2 elongation is due to O$\varepsilon$2···H···O$\delta$2 LBHB is provided by the analysis of another hydrogen bond involving

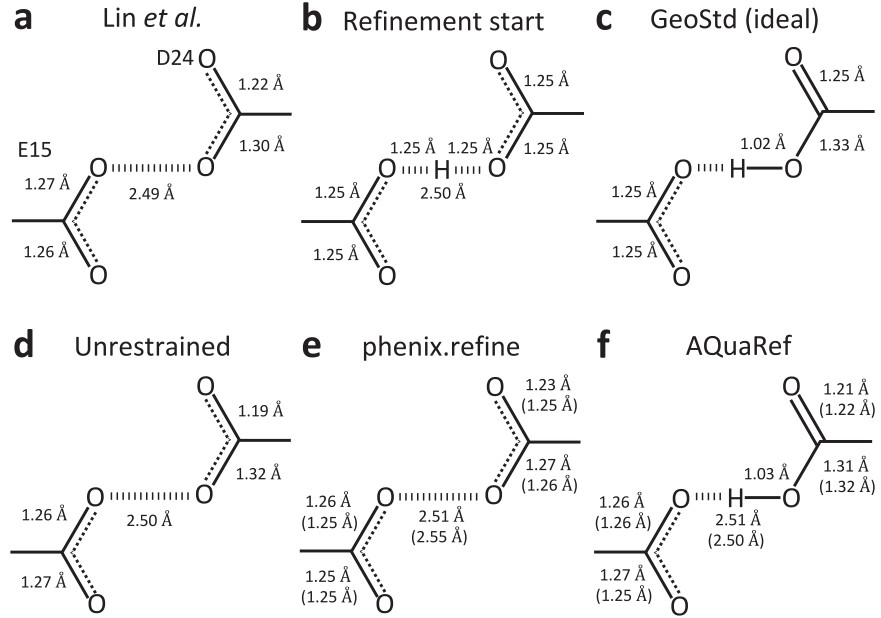

**Fig. 6 | Bond distance analysis in wild-type DJ-1 (PDB code: 5SY6, 1.15 Å).** Bond distances in the moiety of hydrogen bond between Oε2 (E15) and Oδ2 (D24), **a** as measured in downloaded from PDB model, **b** starting geometry for all refinements (H is present only in AQuaRef refinement), **c** ideal library values in Phenix; geometry of –COOH or –COO groups is the same for Asp and Glu residues, **d** unrestrained and **e** restrained refinement with phenix.refine, **f** refinement with AQuaRef. Distances in parentheses correspond to refinement using resolution-truncated data at 2 Å. An H atom is shown only if it was explicitly modelled (present in the PDB model file).

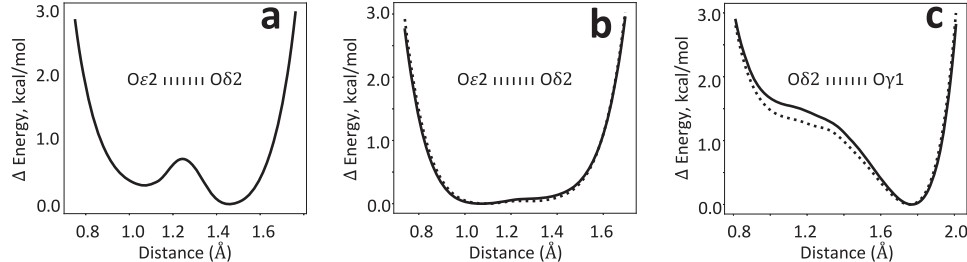

**Fig. 7 | Energy profiles along hydrogen bond.** AIMNet2 energy values relative to their minimum as a function of hydrogen position between corresponding oxygen atoms, **a** Oδ2 (D24) and Oε2 (E15) in DJ-1, **b** Oδ2 (D23) and Oε2 (E14) in YajL, and **c**: Oδ2 (D23) and Oγ1 (T16) in YajL. Solid and dashed lines represent two instances of the bond in the YajL model.

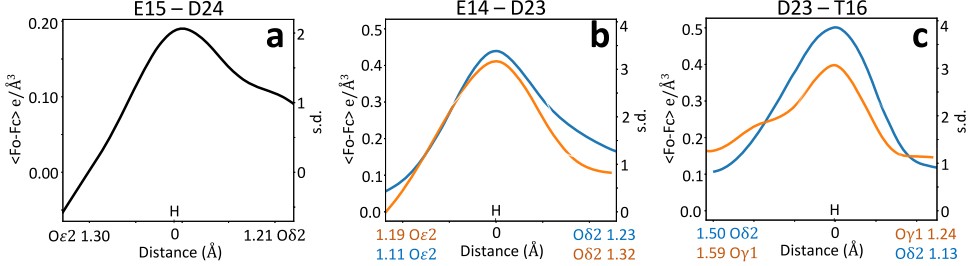

**Fig. 8 | Difference density map along the O-H vector.** Mean values of the difference density map, shown in absolute units (e/Å³) and as standard deviation values along the O-H vector for the analyzed bonds for: **a** DJ-1 and **b**, **c** E. coli YajL models. All peak centers are aligned to the origin. Atoms belonging to chains A and B are shown in blue and orange, correspondingly.

D23 and T16. All three, AQuaRef refinement (Fig. 9d), the AIMNet2 energy profile (Fig. 7c), and difference map density values along the Oδ2 of D23 and Oγ1 of T16 (Fig. 8c), confirm the protonation of T16 and rule out the D23 protonation in the "anti" configuration.

## Discussion

Here, we present AQuaRef, a novel approach to the quantum refinement of entire protein structures, made possible by using ML-accelerated quantum mechanical calculations with AIMNet2. For the

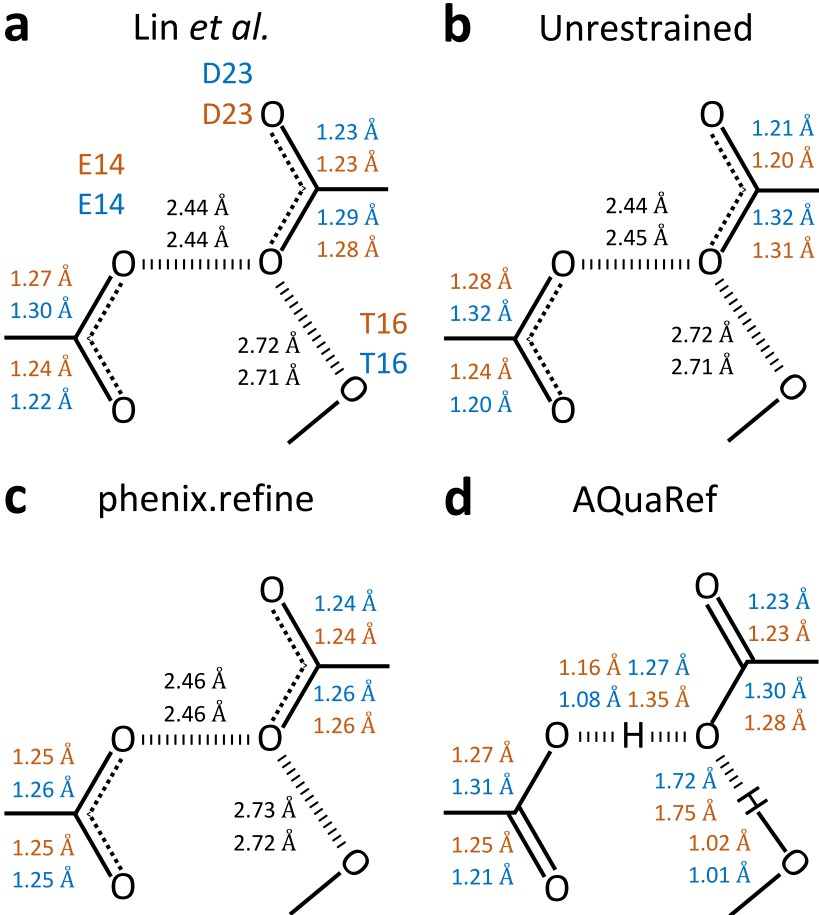

**Fig. 9 | Bond distance analysis in E. coli YajL (PDB code: 5SY4).** Bond distances in the moiety of hydrogen bond between Oε2 (E14) and Oδ2 (D23) across chains A (blue) and B (orange), **a** as measured in downloaded from PDB model, **b** unrestrained and **c** restrained refinement using phenix.refine, **d** refinement with AQuaRef. H atom is shown only if it was explicitly modelled (present in the PDB model file).

first time, this allows for the refinement of full atomic models of realistic protein structures using stereochemical restraints derived from quantum mechanical calculations.

Test refinements using 61 low-resolution X-ray and cryo-EM atomic models show systematic improvements in geometric validation criteria by using QM restraints while maintaining a similar fit to the experimental data and reducing overfitting. The presence of high-resolution homologous atomic models, which are expected to better represent the actual true structures than low-resolution atomic models, allowed us to assess whether these improvements are associated with refined structures becoming closer to the true ones. With a few exceptions (four out of total sixty-one models), atomic models refined with AQuaRef restraints are systematically closer to their high-resolution references. In these four exceptions, the deviation was marginal—less than 1° in torsion angle space as measured by superposition r.m.s.d. This indicates that QM-based refined atomic models not only improve standard validation metrics but also provide more realistic representations of the true structures compared to atomic models refined with standard restraints. Expectedly, refining 10 very high-resolution atomic models did not significantly alter the atomic coordinates but did lead to improved R-factors for all ten models (Supplementary Table 4). The most notable differences compared to

refinement with standard restraints were observed in the position of hydrogen atoms, specifically those with rotational degrees of freedom (Fig. 10a–d), where some of these atoms reoriented during refinement to better fit the data and, at the same time, form favorable hydrogen bonds. Another notable difference is the increased r.m.s. deviations from ideal (library) bond and angle values in the case of AQuaRef refinement (Supplementary Table 4), which, together with improved hydrogen positions is likely to contribute to improved R-factors.

An extended comparison with popular state-of-the-art software packages and refinement methods, including the use of AMBER and Rosetta force fields as refinement restraints, as well as REFMAC5 and Servalcat from the CCP4 software suite, shows that for crystal structure refinement, only Rosetta approaches AQuaRef in terms of the quality of refined atomic model geometries. However, AQuaRef produces slightly improved $R_{free}$ values and significantly better $R_{free}$-$R_{work}$ gaps, indicating reduced data overfitting. It is also worth noting that Rosetta-based refinement is only available for crystal structures using X-ray data, and refinement times with Rosetta are up to an order of magnitude slower. Although Servalcat achieved superior $CC_{mask}$ values compared to AQuaRef (Fig. 5d), this suggests that Servalcat overfits the map, producing higher $CC_{mask}$ values at the cost of significantly poorer model geometry.

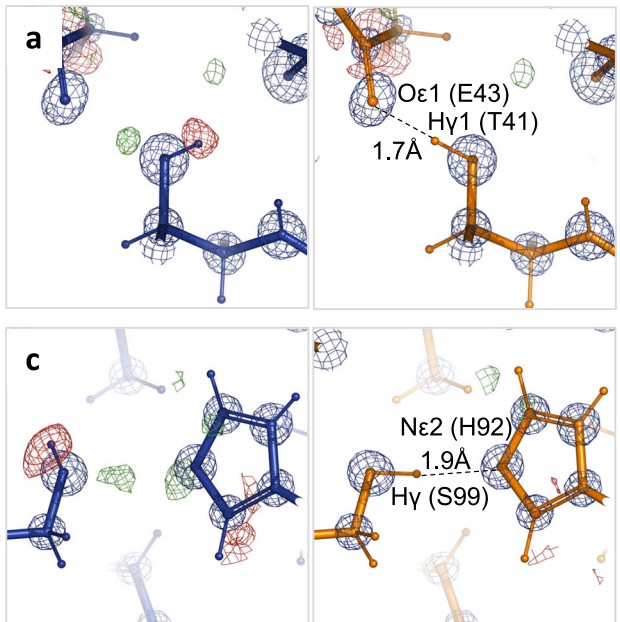

**Fig. 10 | Hydrogen bonds in refined high-resolution models. a–d** Refinement with standard AQuaRef restraints (orange) overlaid with their corresponding 2mFo-DFc and mFo-DFc Fourier maps, contoured at 5σ (blue) and ±2σ (green, red), respectively (PDB 4O8H). The focus is on hydrogen atoms with rotational degrees of freedom that re-orient during refinement with AQuaRef restraints to satisfy the residual map and participate in hydrogen bonding.

The case study of short hydrogen bonds in human DJ-1 and its bacterial homolog YajL, as well as the protonation states of carboxylic acids involved in these hydrogen bonds, highlights the feasibility of AQuaRef in determining proton positions consistent with experimental evidence across diverse scenarios. This process is fully automated and unbiased by the choice of restraints and starting geometry. Additionally, AIMNet2 energy profiles provide further information about the characteristics of hydrogen bonds and protonation states, which can be used to support specific hypotheses.

The method has been implemented in the quantum refinement software (Q|R), which is built upon the CCTBX library[59] and optionally utilizes tools from Phenix. Q|R is accessible within Phenix, thereby making these methods readily available to the broader community of structural biologists.

Currently, AQuaRef is trained using commonly known amino acid residues, which means the method can only be applied to protein-only structures. Another main limitation is that, at present, static disorder (alternate conformations) is not handled in Q|R. Removing both limitations is the subject of future work.

## Methods

### Dataset and AQuaRef model training

Since our goal was the parametrization of ML potential for polypeptides, our training dataset needed to cover chemical (amino acid sequence and protonation states), conformational, and intermolecular degrees of freedom. We began by creating a library of small peptides as SMILES strings. We used all 20 standard amino acids, 11 alternate protonation forms, three options for sequence start (ACE, NH3+, NH2), and four options for the end (NME, NHE, CBX, CBA). We enumerated all possible mono- and di-peptides and selected a random subset for tri- and tetra-peptides. Additionally, we generated SMILES for peptides linked by the cysteine-cysteine disulfide bond and their selenium counterparts. Molecular conformations were generated with OpenEye Omega[60] software using dense torsion sampling. No restrictions were applied to the configurations of the chiral centers, ensuring that the dataset and resulting model should work equally well for D-, L-, and mixed stereochemistry peptides. Intermolecular interactions were modeled by generating intermolecular complexes of 2 to 4 peptides

with random orientations. No prior knowledge of preferred types of secondary structure for polypeptides, naturally occurring amino acid sequences, or experimentally observed intermolecular interactions was used, preventing the data leakage. To manage the size of the dataset and the training process, we limited the size of peptides and complexes to less than 120 atoms, including hydrogens.

Non-equilibrium conformations of peptides and complexes were sampled with molecular dynamics simulations using the GFN-FF[61] force field. Cartesian restraints were added to keep structures near the input structure, with random torsion and intermolecular degrees of freedom. Molecular configurations for labeling (DFT calculations) and inclusion into the training dataset were selected using Query-By-Committee active learning (AL) approach[62]. We started with a random selection of 500k samples, used an ensemble of 4 models, and performed a total of 4 iterations of AL, adding new samples with high uncertainty of energy and atomic forces prediction. In the final iteration of AL, we performed uncertainty-guided optimization of the structures, minimizing the weighted difference of energy prediction and its uncertainty. This type of active sampling finds structures that balance low predicted forces and high energy uncertainty. The entire procedure resulted in a training dataset containing about one million samples, with a median number of 42 atoms per sample.

DFT calculations were performed with the B97M-D4/def2-QZVP[63–66] method using ORCA 5.0.3 software[67]. Since the Q|R does not use periodic boundary conditions, and usually not all ions and solvent molecules are resolved in the refinement, we used implicit treatment of solvent effects with CPCM[68] method using parameters for water as solvent.

The core architecture of the AQuaRef model matches the base AIMNet2 model[36], with few modifications. First, we did not use explicit long-range Coulomb and dispersion interactions, we trained to total DFT-D4 energy instead. With CPCM treatment, the Coulomb term could not be estimated using interactions between partial atomic charges, and also long-range interactions are effectively screened with a polarizable continuum. Long range dispersion interactions beyond the local cutoff of 5 Å have little effect on atomic forces, which are important in Q|R refinement. We also added explicit short-range exponential repulsion term as implemented in GFN1-XTB[69] to make the

potential more robust for the structures with clashes. The model was trained to reproduce the B97M-D4/def2-QZVP energies, forces, and Hirshfeld partial atomic charges. The model was trained starting from random weight initialization for 1.5 million steps with batch size of 256 samples. All training hyper-parameters for the training were preserved from the original AIMNet2 model[36].

## Experimental data and atomic models for test cases

Protein-only, single-conformation high-to-low resolution X-ray crystallography and Cryo-EM models, along with their corresponding experimental datasets, were selected from RCSB and EMDB based on multiple criteria. These criteria include model size (between 1,000 and 10,000 non-hydrogen atoms), resolution (between 2.5 and 4 Å), geometric model quality (MolProbity clashscore better than 50, with no covalent bonds deviating by more than 4 r.m.s.d. from ideal library values), goodness of fit between the model and the experimental data (Cryo-EM: $CC_{mask} > 0.6$, X-ray: $R_{work} < 0.3$), and the availability of a higher-resolution (better than 2 Å) homologous model (main chain superposition r.m.s.d. <1 Å, sequence identity greater than 95%) for each considered model. Additionally, 11 ultra-high resolution single-conformation X-ray models were selected that contained only protein and ordered water atoms.

## Comparison of models

All atoms were used to calculate coordinate r.m.s. deviations between models before and after refinement, as shown in Fig. 3a. Coordinate r.m.s. deviations between models used for test refinements and their high-resolution homologues were calculated using the Phenix tool phenix.superpose_pdbs, which included all non-hydrogen backbone atoms plus Cβ and Cγ atoms where present. R.m.s. deviations in torsion angle space were calculated using CCTBX[62], with matching torsion angles selected as described by Headd et al.[16].

## Atomic model preparation for refinement

Model preparation for refinement (e.g., adding any missing atoms) was done using qr.finalise program of Q|R, which uses the Reduce program[70] to add hydrogen atoms at geometrically predicted positions. Model geometry regularization was done using the Phenix tool phenix.geometry_minimization.

## Model refinement

The exact same input models were used for all trial refinements. Real-space refinement in Phenix was performed using the phenix.real_space_refine program[13]. Four refinement runs were performed independently, starting with the same input maps (cryo-EM) or reflection data (X-ray) and models. The runs included: (1) standard restraints consisting of restraints on bond lengths, bond angles, torsion angles, planes, chirality, and non-bonded repulsion; (2) standard restraints with the addition of secondary-structure restraints; (3) standard restraints with the addition of Ramachandran plot restraints; and (4) standard restraints with the addition of secondary-structure and Ramachandran plot restraints. Reciprocal-space refinement in Phenix was performed using phenix.refine[71] with the exact same four choices of restraints as in real-space refinement.

Quantum-based real- and reciprocal-space refinement was performed using the qr.refine program of Q|R, using all default settings except for the source of QM restraints (AQuaRef).

## Graphics software

Map and model images were prepared using PyMOL[72]. Routine inspection of maps and models was performed using Coot[73]. Plots were generated using Matplotlib[74].

## Reporting summary

Further information on research design is available in the Nature Portfolio Reporting Summary linked to this article.

## Data availability

All data supporting the results of this study can be found in the article, supplementary, source data files and at https://phenix-online.org/phenix_data/afonine/AQuaRef2025/. Refinement parameters are documented in README files, as well as in the Python scripts used to run the refinements. Input data for deposited models were obtained from the Protein Data Bank[75] and Electron Microscopy Data Bank[76], either by using the Phenix tool phenix.fetch_pdb or from the CERES server[77]. The accession codes for 71 structural data used were as follows: 5xb1, 5yi5, 6ezj, 6j6j, 6wik, 6xmx, 6y9w, 6y9x, 7dnj, 7k9i, 7kzn, 7lkh, 7pcq, 7un3, 7vvk, 7vvn, 7vxz, 8aza, 8ckz, 8cl2, 8cl4, 8dat, 8dl8, 8dq0, 8e6k, 8esa, 8fsj, 8g94, 8idn, 8jo4, 8qjx, 8qjy, 8qk3, 8r1f, 8r1g, 8sgj, 8sgt, 8ve0, 8vi2, 8vi4, 8vi5, 1fb5, 1jkt, 1u87, 1wl3, 1xgo, 2fdq, 2yhj, 4xcr, 4yei, 1fp9, 1m10, 1u9o, 1w60, 1×24, 1yab, 2a8z, 2etc, 2h1g, 2jcl, 2pej, 2pnd, 4jp6, 4r5r, 4o8h, 6zm8, 5zgl, 6dkz, 3njw, 2fma, 1tt8 Source data are provided with this paper.

## Code availability

Phenix software is available at: phenix-online.org. Quantum refinement (Q|R) software is available at Github [https://github.com/qrefine/qrefine] and the version used in this study is available at Zenodo [https://doi.org/10.5281/zenodo.16896623]. AQuaRef refinement is available in Phenix starting dev-5395 version.

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

## Acknowledgements
P.V.A., N.W.M., and PDA acknowledge funding from the National Institutes of Health (grants R01GM071939, P01GM063210, and R24GM141254), as well as support from the Phenix Industrial Consortium and the US Department of Energy under Contract No. DE-AC02-05CH11231. OI acknowledges support from the US National Science Foundation (NSF CHE-2154447). AER acknowledges support from the US National Science Foundation (NSF CHE-1802831 and OAC-2311632). MB acknowledges support from the COST Action CA21101 "Confined molecular systems: from a new generation of materials to the stars' (COSY), supported by COST (European Cooperation in Science and Technology) and computer resources provided by Wroclaw Centre for Networking and Supercomputing (http://wcss.pl). This work used Expanse at SDSC and Bridges-2 at PSC through allocation CHE200122 from the Advanced Cyberinfrastructure Coordination Ecosystem: Services & Support (ACCESS) program, which is supported by NSF grants #2138259, #2138286, #2138307, #2137603, and #2138296. This research is part of the Frontera computing project at the Texas Advanced Computing Center. Frontera is made possible by the National Science Foundation award OAC-1818253. This research, in part, was done using resources provided by the Open Science Grid, which is supported by the award 1148698 and the U.S. DOE Office of Science.

## Author contributions
Conceptualization: P.V.A., A.E.R., O.I., and M.B.; Methodology: R.Z., H.G., and K.R.; Software: P.V.A., N.W.M., R.Z., M.P.W., H.K., and B.K.P.; Validation: M.B., K.R., H.G., and P.D.A.; Formal analysis: M.B.; Data Curation: R.Z., H.G., K.R., and P.V.A.; Writing (original draft): P.V.A., M.B., and R.Z.; Writing (review and editing): all authors; Visualization: P.V.A., M.B., H.G., and R.Z.; Supervision, P.V.A., A.E.R., and O.I.; Project administration: P.V.A., A.E.R., O.I., and M.B.; Funding Acquisition: P.D.A., P.V.A., A.E.R., and O.I.

## Competing interests
The authors declare no competing interests.
