## [Transparent Peer Review file · Nature Communications]

AQuaRef: Machine learning accelerated quantum refinement of protein structures

Corresponding Author: Dr Pavel Afonine

Version 1:

Reviewer comments:

Reviewer #1

(Remarks to the Author)

This manuscript describes a new way to more accurately analyze structural results from X-ray crystallographic and CryoEM experiments using quantum mechanical information. The authors have provided good evidence that there are improvements in the resulting atomic models, particularly the hydrogen-bonding networks when compared to classical refinement methods. The fact that the R-free or cryoEM fits are not significantly improved is a bit disappointing but I interpret this to mean that features other than the details accommodated here are still the limiting factors for matching the experimental data, likely conformational variability and solvent structure. Despite this result, I think the method will lead to more usable information in protein structure analysis and thus the method will be impactful, especially when force-field bias is likely to be a factor in the questions being asked.

As for the method, it is good that it is scalable by $O(N)$ and the use of Active Learning to build the training set is clever. The authors are to be commended for making the method easily and broadly available for further testing and evaluation by others.

The paper is clearly written and well illustrated, except that the error bars on the plots are too small to see well and the circles and rhombi are not easy to distinguish without zooming. My only request for changes is to make the figures more legible at the size they are likely to be viewed.

(Remarks on code availability)

Reviewer #2

(Remarks to the Author)

Afonine, Isayev and coworkers reported their development of a new machine learning potentials (MLP), AIMNet2, trained on quantum-mechanical (QM) data, which was combined with a popular refinement program (Phenix) for quantum refinement (QR), so-called AI-enabled Quantum Refinement (AQuaRef), of biomacromolecules. Remarkably, reliable MLP AIMNet2 was employed to refine the entire protein structures from cryo-EM and X-ray crystallography (71 structures) for the first time, apart from one very recent first QR development using ONIOM-type approach and other MLPs (describing the small drug or ligand molecules only). Their systematic evaluation highlights an obvious improvement of geometric quality of the protein structures (such as geometric validation metrics and reduced overfitting) after AQuaRef in an efficient manner by addressing critical limitations of general library-based stereochemical constraints in classical refinements using ML potentials (AIMNet2). This method was also applied to determine protonation states in challenging cases such as DJ-1 and YajL. This impressive QR development in Phenix should be an important contribution for chemistry and biology communities in reliable and efficient refinement of biomacromolecules. I recommend to accept this work for Nature Communications after the authors address the following minor issues.

1. The authors re-trained the AIMNet2 (minor modification of the AIMNet2 architecture) by using specific peptide data, but the accuracy of the re-trained model compared to the reference ω B97M-D3 method is not shown in the manuscript. The authors should give more information in the supplementary material. Also, the authors should give more comments to explain why an implicit water solvent model was used to train AIMNet2 in this work? Although AIMNet2 is a new and more reliable MLP,

development of MLPs typically consider short-range interactions with adjacent atoms only. Therefore, MLPs may not be able to get higher polarization/strength of amide hydrogen bonds enhanced by many-body interactions/effects. Could the authors also give some comments on this point. Also, the reference for AIMNet2 should be updated by their just online Chem. Sci. paper (doi.org/10.1039/D4SC08572H).

2. Timeline given in Fig. 1 only includes very few limited method developments, which are mainly relevant to the Phenix team and might give incomplete/misleading picture for general readers. I recommend the authors to either add a few words pointing to the Phenix development or add much more QR developments from the other teams. For the first point, it could still be possible to use highly efficient RI + DFT methods (esp. pure DFT functionals) for treating proteins with ~200-300 atoms using Gaussian-based basis sets and even larger-size proteins using plane-wave basis sets in year 2010. In addition, a few fragmentation-based or linear-scale QM methods were also developed.

3. Page 4, "recent" and "ONIOM-like" should be changed by "recent novel" and "multi-layer ONIOM-type", respectively. Apart from many citations for the Phenix and QR program development. The authors can also consider to cite a few recent new QR methods using different approaches and one classical refinement paper, e.g. IUCrJ (2024). 11, 921; J. Chem. Theory Comput. 2021, 17, 3783; Acta Crystallogr A 1978, 34, 931.

4. Page 6, it should be helpful for many general readers of Nature Communications, if the authors briefly define O(N) or related terminologies once.

5. Fig. 3a shows the performance on Cryo-EM and X-ray models. As shown in Supplementary Tables 1 and 3, the improvement (difference in mean values between R4 and R1) for Cryo-EM and X-ray methods are quite different (Molprobit score: -1.61 vs -1.01; Rama-Z: 4.63 vs 1.90; CaBLAM disf.: -8.77 vs -2.66; CaBLAM outl.: -3.30 vs -1.29; RMSD: -0.03 vs 0.13), it is recommended to separate the Cryo-EM and X-ray models in Fig. 3a.

6. Page 10, it should be helpful for the general readers, if the authors explicitly mention numbers of the exceptions.

7. As always, accurate determination of hydrogen positions of biomacromolecules by X-ray and Cryo-EM have long been very challenging, esp. for very strong hydrogen bonds (e.g. highly covalent feature for low barrier hydrogen bond, LBHB), by using the X-ray and Cryo-EM experimental data or computational chemistry. Neutron diffraction is the common/better way to determine the hydrogen positions instead. As to the computational chemistry, apart from a reliable method to give its potential energy, two quantum effects (zero-point energy correction and quantum tunnelling) should be included. The observed very short O--O distances (~2.5 Ang.) between the two carboxylic (D/E) groups in DJ-1 and YajL should give a strong implication of a very strong H-bond. We feel that all except the strong hydrogen bond section is very clear and enough for Nature Communications. An inclusion of many results for the strong hydrogen bond might lead to some uncertainties. If the authors prefer to highlight this part, they should use AIMNet2 to apply any known small organic molecules containing LBHB (e.g. formic-acid dimer) from literatures or compare their energy curves obtained by any highly-reliable method (e.g. CCSD(T)). Alternatively, the authors can select one protein system with neutron diffraction or solid-state NMR data (as independent experimental support) for additional QR simulations to accurately determine and compare the challenging proton positions. Anyway, the current AQuaRef results are the most reasonable. However, I wonder how these distance change if the anionic state of the two carboxylic groups is considered after AQuaRef (not a must to do). Moreover, for line 339 or 346, please add/define LBHB abbreviation for 'Low Barrier Hydrogen Bond' once. Furthermore, it is quite confusing to state that both D23 and E14 are protonated (line 336): both are neutral. Either D23 or E14 is protonated instead.

8. Line 295, the resolution of PDB code for 5SY6 is "1.15" Ang., but "1.1" Ang. was used in the main text. In addition, as "2 Ang." was given in the Fig 5 caption, it should be good to add "1.15 Ang" after "5SY6" in Fig. 5 caption as the comparison.

9. In Supplementary Table 6, what type of the computational hardware was used to obtain the refinement time? Still a single Nvidia H100 PCIE 80GB GPU?

10. Refs. 22 and 56 are identical. Ref. 42 should be fixed. Ref. 71: "1 Edited by J. Thornton" should be deleted.

(Remarks on code availability)

I briefly browsed the qrefine code website, which seemingly provide instructions (README file) to run this code

Reviewer #3

(Remarks to the Author)

(Remarks on code availability)

Reviewer #4

(Remarks to the Author)

This manuscript presents AQuaRef, an AI-enabled quantum refinement method based on AIMNet2 neural network potential.

Unlike standard approaches that use library-based stereochemical restraints, quantum refinement exploits restraints derived from quantum-mechanical calculations for the specific macromolecule in consideration. On a dataset of 41 cryo-EM and 30 X-ray structures, the method seems to fare well compared to standard approaches.

I have the following comments:

1. The core architecture of the AQuaRef model is based on the AIMNet2 model, which seems to be recently published (<https://pubs.rsc.org/en/content/articlelanding/2025/sc/d4sc08572h>), although the authors here still cited a preprint <https://doi.org/10.26434/chemrxiv-2023-296ch-v2>. This somewhat diminishes the novelty of this manuscript.
2. AQuaRef heavily relies on AIMNet2, yet AIMNet2 neural network model and more importantly the modifications made in AIMNet2 as part of AQuaRef are not well articulated in this manuscript.
3. Method details are often so sparse that it is impossible to figure out the details. For example, what was the detailed neural architecture of AIMNet2, how the hyperparameters were tuned, how the model was recalibrated to exclude explicit long-range Coulomb and dispersion interactions, how explicit short-range exponential repulsion terms were added, and so on, remain unclear.
4. What measures, if any, were taken to data leakage on AIMNet2 model training and performance evaluation 41 cryo-EM and 30 X-ray structures? Did the authors use both sequence and structural homology cutoff?
5. It might be useful to demonstrate the effectiveness of the Query-By-Committee active learning (AL) approach, perhaps by conducting additional ablation study.
6. In Fig. 3a, AQuaRef (orange) seems to have much wider distribution of r.m.s. deviation of refined model from initial model compared to the baseline. I am concerned about the model-to-data fit for the structures that are beyond one standard deviation of the mean.

(Remarks on code availability)

Version 2:

Reviewer comments:

Reviewer #1

(Remarks to the Author)

Nice revision. Good to go as far as I am concerned.

(Remarks on code availability)

Reviewer #2

(Remarks to the Author)

The authors have addressed all of our comments. I would recommend to accept this work. The only very minor and last comment is to add "some" or "selected" before "key milestones" in the Fig 1 caption, as some key QR works were not included in Fig. 1. I do not need to review this work more.

(Remarks on code availability)

Reviewer #3

(Remarks to the Author)

(Remarks on code availability)

Reviewer #4

(Remarks to the Author)

My comments have been addressed.

(Remarks on code availability)

Reviewer #1:

This manuscript describes a new way to more accurately analyze structural results from X-ray crystallographic and CryoEM experiments using quantum mechanical information. The authors have provided good evidence that there are improvements in the resulting atomic models, particularly the hydrogen-bonding networks when compared to classical refinement methods. The fact that the R-free or cryoEM fits are not significantly improved is a bit disappointing but I interpret this to mean that features other than the details accommodated here are still the limiting factors for matching the experimental data, likely conformational variability and solvent structure. Despite this result, I think the method will lead to more usable information in protein structure analysis and thus the method will be impactful, especially when force-field bias is likely to be a factor in the questions being asked.

We thank the reviewer for their thoughtful comments and positive assessment of our work. As noted, our method focuses on improving the geometric accuracy of atomic models by incorporating quantum mechanical information, particularly enhancing features such as hydrogen-bonding networks.

Regarding the model-to-data fit metrics: for test refinements performed at lower resolution, we did not anticipate substantial changes in global metrics such as R-factors (crystallography) or CCmask (cryoEM), as the magnitude of coordinate adjustments introduced by AQuaRef falls below the resolution limit of the data in most cases. However, in high-resolution refinements, AQuaRef consistently yields improved R-factors compared to those obtained with Phenix, as shown in Table 5 (High-resolution X-ray refinement, Supplementary Data). These improvements are also evident when comparing to recomputed values from the original PDB-deposited models.

As the reviewer rightly points out, limitations such as bulk solvent modeling and conformational heterogeneity remain significant barriers to achieving closer agreement with experimental data. Our method addresses one important component — accurate atomic geometries and interactions, but we agree that other structural features continue to impact overall model quality and data fit.

As for the method, it is good that it is scalable by $O(N)$ and the use of Active Learning to build the training set is clever. The authors are to be commended for making the method easily and broadly available for further testing and evaluation by others.

It is the authors' intention—and a long-term development goal—to establish AQuaRef as a standard step in the structure solution pipeline for crystallography and cryo-EM, widely accepted by the structural biology community.

The paper is clearly written and well illustrated, except that the error bars on the plots are too small to see well and the circles and rhombi are not easy to distinguish without zooming. My only request for changes is to make the figures more legible at the size they are likely to be viewed.

We have revised Figure 3 to improve legibility. Specifically, we increased and standardized the font sizes for all textual elements across all panels, enlarged the markers in panel (f), and

increased the size of the bars. We hope these changes will make the figure easier to interpret at typical viewing scales, as suggested by the reviewer.

While revising Figure 3, we also realized that incorrect data columns were used to plot panel 3e. This has now been corrected. Please note that this correction does not affect the interpretation or conclusions drawn from Figure 3e.

Reviewer #2:

Afonine, Isayev and coworkers reported their development of a new machine learning potentials (MLP), AIMNet2, trained on quantum-mechanical (QM) data, which was combined with a popular refinement program (Phenix) for quantum refinement (QR), so-called AI-enabled Quantum Refinement (AQuaRef), of biomacromolecules. Remarkably, reliable MLP AIMNet2 was employed to refine the entire protein structures from cryo-EM and X-ray crystallography (71 structures) for the first time, apart from one very recent first QR development using ONIOM-type approach and other MLPs (describing the small drug or ligand molecules only). Their systematic evaluation highlights an obvious improvement of geometric quality of the protein structures (such as geometric validation metrics and reduced overfitting) after AQuaRef in an efficient manner by addressing critical limitations of general library-based stereochemical constraints in classical refinements using ML potentials (AIMNet2). This method was also applied to determine protonation states in challenging cases such as DJ-1 and YajL. This impressive QR development in Phenix should be important contribution for chemistry and biology communities in reliable and efficient refinement of biomacromolecules. I recommend to accept this work for Nature Communications after the authors address the following minor issues.

We sincerely thank the reviewer for their positive and encouraging assessment of our work. We are pleased that the reviewer recognized the novelty and potential impact of integrating AIMNet2 with Phenix for quantum refinement of biomacromolecular structures. We appreciate their support of our systematic evaluation and the broader implications for structural biology and chemistry.

1. The authors re-trained the AIMNet2 (minor modification of the AIMNet2 architecture) by using specific peptide data, but the accuracy of the re-trained model compared to the reference ω B97M-D3 method is not shown in the manuscript. The authors should give more information in the supplementary material. Also, the authors should give more comments to explain why an implicit water solvent model was used to train AIMNet2 in this work?

We apologize for the confusion and absence of a clear description of AIMNet2 throughout the paper. We extended the Introduction and Methods sections with details about the AIMNet2 model.

Although AIMNet2 is a new and more reliable MLP, development of MLPs typically consider short-range interactions with adjacent atoms only. Therefore, MLPs may not be able to get higher polarization/strength of amide hydrogen bonds enhanced by many-body interactions/effects. Could the authors also give some comments on this point.

The original AIMNet2 model includes explicit long-range Coulomb interactions as a sum of pairwise interactions between atomic point charges. However, with the CPCM solvation model, the Coulomb term also includes interactions with a cloud of added charges on the molecular surface. This term is very difficult to include explicitly. Therefore, we adopted the approach that is currently the standard for ML interatomic potentials: including Coulomb interaction implicitly with the message passing mechanism. The choice for implicit dispersion interactions was also motivated by computational performance considerations to preserve strictly linear $O(N)$ behavior.

The AQueRef model cannot truly account for all types of long-range interactions. However, each atom has an explicit receptive field of 5\AA and an implicit range up to 15\AA due to 3 rounds of message passing. This is enough to describe a majority of the effects that are relevant to influence on molecular geometry.

Also, the reference for AIMNet2 should be updated by their just online Chem. Sci. paper (doi.org/10.1039/D4SC08572H).

Thank you for the note. The reference was updated.

2. Timeline given in Fig. 1 only includes very few limited method developments, which are mainly relevant to the Phenix team and might give incomplete/misleading picture for general readers. I recommend the authors to either add a few words pointing to the Phenix development or add much more QR developments from the other teams. For the first point, it could still be possible to use highly efficient RI + DFT methods (esp. pure DFT functionals) for treating proteins with ~ 200 - 300 atoms using Gaussian-based basis sets and even larger-size proteins using plane-wave basis sets in year 2010. In addition, a few fragmentation-based or linear-scale QM methods were also developed.

We agree that Figure 1 may have given the impression of representing a comprehensive timeline of quantum refinement developments, which was not our intent. The figure is meant to highlight selected milestones specifically relevant to the use of quantum mechanical methods as sources of geometric restraints in the refinement of biomacromolecular structures. This intent is now clarified in the revised figure caption and main text.

We also note that only one of the milestones shown—the third, involving the use of fragmentation-based quantum refinement—is directly related to the Phenix/QR project. This step marked a key advance by enabling full-protein refinement against cryo-EM or X-ray data using QM-derived restraints.

Regarding the use of DFT methods around 2010: while single-point quantum calculations on systems of ~ 200 – 300 atoms may have been feasible at that time, the application of such methods to full structure refinement workflows—which require thousands of energy and gradient evaluations—was prohibited by computational cost. As such, these approaches were not yet

practical for routine refinement of full protein structures that typically consist of tens to hundreds of thousands of atoms.

3. Page 4, “recent” and “ONIOM-like” should be changed by “recent novel” and “multi-layer ONIOM-type”, respectively. Apart from many citations for the Phenix and Q|R program development. The authors can also consider to cite a few recent new QR methods using different approaches and one classical refinement paper, e.g. IUCrJ (2024). 11, 921; J. Chem. Theory Comput. 2021, 17, 3783; Acta Crystallogr A 1978, 34, 931.

We have revised the manuscript to incorporate all these suggestions.

4. Page 6, it should be helpful for many general readers of Nature Communications, if the authors briefly define $O(N)$ or related terminologies once.

Big-O notation is commonly used in computer science to describe an algorithm's performance changes as the problem size grows. Rather than measuring the exact time or memory usage, it provides a high-level classification of an algorithm's scaling in its worst-case scenario. It provides developers with a simple, standardized method for comparing the efficiency of different algorithms and predicting their performance at a large scale.

We clarified the meaning of $O(N)$ in the revised manuscript.

5. Fig. 3a shows the performance on Cryo-EM and X-ray models. As shown in Supplementary Tables 1 and 3, the improvement (difference in mean values between R4 and R1) for Cryo-EM and X-ray methods are quite different (Molprobit score: -1.61 vs -1.01; Rama-Z: 4.63 vs 1.90; CaBLAM disf.: -8.77 vs -2.66; CaBLAM outl.: -3.30 vs -1.29; RMSD: -0.03 vs 0.13), it is recommended to separate the Cryo-EM and X-ray models in Fig. 3a.

We appreciate the reviewer's suggestion to separate the cryo-EM and X-ray models in Figure 3a. Our original intention was indeed to present these datasets separately. However, during figure preparation, we found that doing so would substantially increase the visual complexity of an already information-dense figure, potentially reducing its clarity for readers. In fact, Reviewer #1 specifically recommended improving the legibility of this figure.

While we acknowledge that the improvements differ between cryo-EM and X-ray models—as shown in Supplementary Tables 1 and 3—our aim in Figure 3a was to highlight the overall trend in improvements resulting from the use of AQuaRef across all models, rather than to focus on comparisons between experimental techniques.

That said, the full breakdown of results by experimental method is provided in the Supplementary Materials, allowing interested readers to explore these distinctions in detail—as the reviewer has done. For these reasons, we propose to keep the content of Figure 3a unchanged (aside from enhancing its legibility as suggested by Reviewer #1).

6. Page 10, it should be helpful for the general readers, if the authors explicitly mention numbers of the exceptions.

The revised manuscript now specifies the number of exceptions, as well as the magnitude of the differences.

7. As always, accurate determination of hydrogen positions of biomacromolecules by X-ray and Cryo-EM have long been very challenging, esp. for very strong hydrogen bonds (e.g. highly covalent feature for low barrier hydrogen bond, LBHB), by using the X-ray and Cryo-EM experimental data or computational chemistry. Neutron diffraction is the common/better way to determine the hydrogen positions instead. As to the computational chemistry, apart from a reliable method to give its potential energy, two quantum effects (zero-point energy correction and quantum tunnelling) should be included. The observed very short O--O distances (~2.5 Ang.) between the two carboxylic (D/E) groups in DJ-1 and YajL should give a strong implication of a very strong H-bond. We feel that all except the strong hydrogen bond section is very clear and enough for Nature Communications. An inclusion of many results for the strong hydrogen bond might lead to some uncertainties. If the authors prefer to highlight this part, they should use AIMNet2 to apply any known small organic molecules containing LBHB (e.g. formic-acid dimer) from literatures or compare their energy curves obtained by any highly-reliable method (e.g. CCSD(T)). Alternatively, the authors can select one protein system with neutron diffraction or solid-state NMR data (as independent experimental support) for additional QR simulations to accurately determine and compare the challenging proton positions. Anyway, the current AQuaRef results are the most reasonable. However, I wonder how these distance change if the anionic state of the two carboxylic groups is considered after AQuaRef (not a must to do).

We thank the reviewer for their thoughtful and insightful comments regarding the challenges in accurately determining hydrogen positions, particularly in the context of low-barrier hydrogen bonds (LBHBs). We fully agree that this remains a complex area, and that techniques such as neutron diffraction and solid-state NMR are better suited to definitively resolving hydrogen positions, except for scenarios similar to discussed here: <https://doi.org/10.1002/pro.3716> .

This section was included at the request of the editor to provide a case study demonstrating how AQuaRef can capture structural features not easily accessible through conventional refinement approaches. In particular, one strength of AQuaRef is its ability to describe proton transfer phenomena without requiring predefined protonation states—a limitation of most current refinement methods.

We selected the DJ-1 and YajL systems due to their reported relevance to LBHBs and their availability in high-resolution structures, which allowed us to explore AQuaRef's performance with respect to the unrestrained refinement relying solely on the experimental data in modeling such cases. While we recognize that the absence of independent experimental validation (e.g. from neutron diffraction or NMR) limits definitive conclusions, our aim was not to prove the presence of LBHBs per se, but rather to illustrate AQuaRef's capacity to model plausible

protonation states in line with the evidence from the experimental electron density and energetics, and without introducing biasing assumptions.

We appreciate the reviewer's suggestions for future benchmarking using well-characterized systems or structures solved using neutron data. We agree that such studies would be valuable for further validating AQuaRef, and we consider this an important direction for future work.

Moreover, for line 339 or 346, please add/define LBHB abbreviation for 'Low Barrier Hydrogen Bond' once.

We have added LBHB abbreviation in the first instance.

Furthermore, it is quite confusing to state that both D23 and E14 are protonated (line 336): both are neutral. Either D23 or E14 is protonated instead.

We thank the reviewer for pointing out this ambiguity. Our intention was to convey that the proton appears to be shared between E14 and D23, consistent with a low-barrier hydrogen bond. This is supported by the observation that each residue exhibits one C–O bond distance close to 1.3 Å and the other around 1.2 Å (Figure 8d), suggesting partial delocalization of the proton. We have clarified this point in the revised manuscript.

8. Line 295, the resolution of PDB code for 5SY6 is "1.15" Ang., but "1.1" Ang. was used in the main text. In addition, as "2 Ang." was given in the Fig 5 caption, it should be good to add "1.15 Ang" after "5SY6" in Fig. 5 caption as the comparison.

We have implemented requested correction in the revised manuscript.

9. In Supplementary Table 6, what type of the computational hardware was used to obtain the refinement time? Still a single Nvidia H100 PCIe 80GB GPU?

All refinement timings reported in Supplementary Table 6 were obtained using a system equipped with three Nvidia A100 PCIe 40 GB GPUs. However, only one card was used for all refinements presented in this study.

10. Refs. 22 and 56 are identical. Ref. 42 should be fixed. Ref. 71: "1 Edited by J. Thornton" should be deleted.

We have removed duplicated reference 22/56 (Amber in Phenix) and fixed references 42 and 71.

I briefly browsed the qrefine code website, which seemingly provide instructions (README file) to run this code

In addition to the README and usage instructions provided at qrefine web site, detailed documentation for AQuaRef is also available through the official Phenix documentation at the following link:

Reviewer #3:

We thank Early Career Researcher for co-review and appreciate Nature Communication initiative.

Reviewer #4:

This manuscript presents AQuaRef, an AI-enabled quantum refinement method based on AIMNet2 neural network potential. Unlike standard approaches that use library-based stereochemical restraints, quantum refinement exploits restraints derived from quantum-mechanical calculations for the specific macromolecule in consideration. On a dataset of 41 cryo-EM and 30 X-ray structures, the method seems to fare well compared to standard approaches.

We thank the reviewer for their positive feedback on our work.

1. The core architecture of the AQuaRef model is based on the AIMNet2 model, which seems to be recently published (<https://pubs.rsc.org/en/content/articlelanding/2025/sc/d4sc08572h>), although the authors here still cited a preprint <https://doi.org/10.26434/chemrxiv-2023-296ch-v2>. This somewhat diminishes the novelty of this manuscript.

The focus and novelty of the work presented here are not limited to AIMNet2 alone, but rather the full-protein atomic model refinement using QM-derived geometric restraints, aided by AI/ML. While AIMNet2 is an important component of our approach and is now formally published, the primary focus of our work is its application to full-protein atomic model refinement using QM-derived geometric restraints within the AQuaRef framework. To the best of our knowledge, this represents the first demonstration of quantum refinement of entire biomacromolecules using machine learning potentials, extending beyond previous efforts that were limited to small molecules or local regions using mixed QM and MM potentials. We have updated the citation to reference the peer-reviewed AIMNet2 publication and clarified this distinction in the revised manuscript.

2. AQuaRef heavily relies on AIMNet2, yet AIMNet2 neural network model and more importantly the modifications made in AIMNet2 as part of AQuaRef are not well articulated in this manuscript.

We apologize for the confusion and absence of a clear description of AIMNet2 throughout the paper. We extended the main text (Methods).

3. Method details are often so sparse that it is impossible to figure out the details. For example, what was the detailed neural architecture of AIMNet2, how the hyperparameters were tuned, how the model was recalibrated to exclude explicit long-range Coulomb and dispersion interactions, how explicit short-range exponential repulsion terms were added, and so on, remain unclear.

We used the same architecture of the AIMNet2 model, which is described in detail in the original paper [<https://doi.org/10.1039/D4SC08572H>]. No further hyperparameter tuning was performed. No recalibration was needed to exclude explicit long-range Coulomb and dispersion interactions. We thank the reviewer for pointing out that the details of the short-range repulsion term are indeed missing in the manuscript. We added the following text and the reference to address this:

We also added explicit short-range exponential repulsion term as implemented in GFN1-xTB model (10.1021/acs.jcim.1c00432) to make the potential more robust for the structures with clashes.

4. What measures, if any, were taken to data leakage on AIMNet2 model training and performance evaluation 41 cryo-EM and 30 X-ray structures? Did the authors use both sequence and structural homology cutoff?

We prevented data leakage by training the model on random short peptides, using no information about naturally occurring amino acid sequences or experimentally observed intermolecular interactions.

5. It might be useful to demonstrate the effectiveness of the Query-By-Committee active learning (AL) approach, perhaps by conducting additional ablation study.

We apologize for this confusion. The general effectiveness of this approach has been demonstrated in the literature before. For demonstration of the effectiveness of query-by-committee approach, please refer to our previous publication (<https://doi.org/10.1063/1.5023802>). As our goal is to develop an ML potential which can be used for protein structure refinement (AQuaRef framework), we do not make claims about the data efficiency of our approach to construct a training dataset for the AQuaRef model.

6. In Fig. 3a, AQuaRef (orange) seems to have much wider distribution of r.m.s. deviation of refined model from initial model compared to the baseline. I am concerned about the model-to-data fit for the structures that are beyond one standard deviation of the mean.

The convergence radius for reciprocal space refinement is around 0.8 Å (Acta Cryst. (1978), A34, 791-809), while for real space refinement, it is somewhat larger – around 1–2 Å (Acta Cryst. (2018). D74, 531–544)—due to the specific refinement target function used. The deviations observed between the refined and starting models in Figure 3a fall well within these expected ranges, especially considering that the initial models had relatively poor geometric quality, which may have required larger adjustments.